# Biochemical Characterisation of Human Transglutaminase 4

**DOI:** 10.3390/ijms222212448

**Published:** 2021-11-18

**Authors:** Zsuzsa Csobán-Szabó, Bálint Bécsi, Saïd El Alaoui, László Fésüs, Ilma Rita Korponay-Szabó, Róbert Király

**Affiliations:** 1Department of Biochemistry and Molecular Biology, Faculty of Medicine, University of Debrecen, 4032 Debrecen, Hungary; szabo.zsuzsa@med.unideb.hu (Z.C.-S.); fesus@med.unideb.hu (L.F.); 2Molecular Cell and Immunobiology Doctoral School, University of Debrecen, 4032 Debrecen, Hungary; 3Department of Medical Chemistry, Faculty of Medicine, University of Debrecen, 4032 Debrecen, Hungary; bbalint@med.unideb.hu; 4Research Department, Covalab S.A.S., 69500 Bron, France; elalaoui@covalab.com; 5Department of Pediatrics, Faculty of Medicine, University of Debrecen, 4032 Debrecen, Hungary; ilma.korponay-szabo@tuni.fi

**Keywords:** transglutaminase, TG4, TGp, protein crosslinking, enzyme activity, substrate search, proteomic analysis, tissue distribution, database reanalysis, prostate cancer

## Abstract

Transglutaminases are protein-modifying enzymes involved in physiological and pathological processes with potent therapeutic possibilities. Human TG4, also called prostate transglutaminase, is involved in the development of autoimmune and tumour diseases. Although rodent TG4 is well characterised, biochemical characteristics of human TG4 that could help th e understanding of its way of action are not published. First, we analysed proteomics databases and found that TG4 protein is present in human tissues beyond the prostate. Then, we studied in vitro the transamidase activity of human TG4 and its regulation using the microtitre plate method. Human TG4 has low transamidase activity which prefers slightly acidic pH and a reducing environment. It is enhanced by submicellar concentrations of SDS suggesting that membrane proximity is an important regulatory event. Human TG4 does not bind GTP as tested by GTP-agarose and BODIPY-FL-GTPγS binding, and its proteolytic activation by dispase or when expressed in AD-293 cells was not observed either. We identified several potential human TG4 glutamine donor substrates in the AD-293 cell extract by biotin-pentylamine incorporation and mass spectrometry. Several of these potential substrates are involved in cell–cell interaction, adhesion and proliferation, suggesting that human TG4 could become an anticancer therapeutic target.

## 1. Introduction

Transglutaminases are protein modifying enzymes sharing a Cys-His-Asp catalytic triad [1]. Their most prevalent enzymatic activity is the Ca^2+^-dependent formation of N^ε^(γ-glutamyl)lysine isopeptide bond between glutamine and lysine residues of proteins or incorporation of biogenic amines into protein’s glutamine residues [2]. This crosslinking activity plays an essential role in the clotting of body fluids by secreted transglutaminases, Factor XIII-A (FXIII-A) [3] and prostate-specific transglutaminase (transglutaminase type 4, TG4, TGp), which are involved in two analogous biological processes, blood clotting and copulatory plug formation [4], respectively. While blood clotting Factor XIII-A is a well-characterised enzyme, TG4 is, however, a less studied member of the transglutaminase enzyme family.

In rodents, TG4 is highly expressed in the coagulating gland and contributes to the formation of the copulatory plug, which is an insoluble protein polymer formed upon ejaculation of the seminal fluid in the vagina [4]. In TG4-/- mice, the copulatory plug formation is missing, and this is associated with decreased litter size and reduced fertility [5]. TG4 also may be involved in the suppression of the antigenicity of sperm cells because inhibition of transglutaminase activity in rabbit prostatic fluid restores the lymphocyte stimulation by the spermatozoa, which was also confirmed in a rat model [6,7]. Although TG4 is called prostate-specific transglutaminase, it is also expressed in mouse aortic smooth muscle cells and vena cava [8]. Rat and mouse TG4 are well characterised from purified enzymes from the coagulating glands or from cloned proteins [9,10,11,12,13,14], but our knowledge about human TG4 is feeble.

Human TG4 (hTG4) is expressed in the prostate and present in the seminal fluid, but there is no copulatory plug formation in humans. Human TG4 may still regulate semen viscosity and the maturation and immunogenicity of sperm cells covalently modifying proteins by its transglutaminase activity [15,16]. hTG4 is also present in the saliva [17] and in the vesicular fraction of urine [18]. Human TG4 is frequently expressed by breast and prostate cancer cells, in which it is an adverse prognostic marker associated with higher invasiveness [19,20]. hTG4 is an autoantigen in autoimmune polyendocrine syndrome type 1, and contributes to the development of prostatitis resulting in male infertility [21]. These presumed roles of hTG4 look controversial since evolutional biology studies claim that the gene of TG4 (TGM4) is dead, has lost its biological function due to the lack of copulatory plug formation [22] and its high polymorphism in humans [23] indicates low evolutional pressure and dispensability. Indeed, there is a lack of both detailed biochemical characterisation and demonstration of the biological significance of hTG4 in physiological and pathological processes.

Our study aimed to characterise the enzymatic properties of hTG4 experimentally, as well as to collect and analyse the publicly available human-related expression data at the protein level. Our experiments and proteomics database analyses show that human TG4 is expressed in various parts of the gastrointestinal tract in addition to the male genital organs. hTG4 has a low catalytic transglutaminase activity preferring slightly acidic pH and reducing conditions and sodium dodecyl sulphate (SDS) can enhance it. Human TG4 has lost its GTP binding property and is potentially being under evolutional changes. Potential cellular substrates identified in this study support the theory that hTG4 can modulate cell adhesion and migration, influencing cancer invasiveness.

## 2. Results

### 2.1. Analysis of Proteomics Datasets for the Expression of hTG4 Protein in the Human Body

In order to collect evidence about the tissue expression of hTG4 databases were checked. The Human Protein Atlas was created based on immunohistochemistry, and the applied antibodies are roughly validated based on consistency with the RNA expression data in various tissues [24]. According to this database, hTG4 is expressed in the prostate and skeletal muscle at the protein level under physiological conditions. In addition, it was also present in one breast cancer sample showing strong staining and cytoplasmic/membranous localisation. In prostate cancers, the presence of hTG4 also was detected several times with various staining intensities, mostly having cytoplasmic and membranous localisation, but hTG4 is not considered as a prognostic marker of these tumours. The appearance and spread of high sensitivity mass spectrometers opened another approach to screen protein expressions.

In the Uniprot database, hTG4 is linked to several proteomics databases, of which MassIVE [25], Peptide Atlas [26] and Proteomics DB [27] provide the possibility to reach detailed experimental settings and gained metadata of each observed hTG4 peptides in various projects. By reviewing these databases and analysing the metadata of each observed hTG4 peptide, experimental conditions and sample origin, we collected 41 tissues where peptide fragments of hTG4 were detected (Appendix A).

Due to sequence similarity between various proteins, particularly within the transglutaminase enzyme family, there are observed peptides that are shared between proteins. The YPEGSSEER peptide or its post-translationally modified forms are very commonly detected, but it also presents in the hTG2 sequence. In many tissues, hTG4 was detected only based on this one shared peptide. After exclusion of these nonunique peptides, the presence of hTG4 could be confirmed by unique peptides only in nine tissues (Table 1). These data support that, besides the male genital tract, hTG4 is present physiologically in the foetal heart, spleen, salivary glands, colon and urinary bladder.

### 2.2. hTG4 Has Curiously Low Transamidase Activity in the Transglutaminase Family

To characterise hTG4 experimentally, we produced recombinant hTG4 and tested its transamidase activity. Since there are no data on the best hTG4 substrates, the widely used so-called microtitre plate biotin-pentylamine incorporation assay was used. The transamidase activity of 0.5 µg hTG4 was 0.62 ± 0.29 mAbs/min, which is extremely low compared to the positive control recombinant human TG2 (29.25 ± 5.93 mAbs/min; at 5 mM [Ca^2+^]). The transamidase activity of commercial recombinant hTG4 (Zedira) was similar to the recombinant enzyme produced in our laboratory. Still, when the effect of increasing amounts of hTG4 was measured on the transamidase activity, it demonstrated a good linear correlation (Figure 1A) reaching 7.31 ± 0.33 mAbs/min at 4 µg protein. To further confirm the validity of the measurement of transamidase activity, the Ca^2+^-dependence was also tested (Figure 1B). hTG4 activity demonstrated an optimal [Ca^2+^] at 5 mM. A higher Ca^2+^ level (10 mM) showed an inhibitory effect on the activity, probably due to the aggregation of the protein.

The Zedira GmBH (Darmstadt, Germany) sells recombinant human transglutaminases and shares the transamidase activities of these proteins measured by a kinetic dansyl cadaverine incorporation assay (Table 2). These activity values support our observation and confirm that hTG4 has a very low transamidase activity compared to the other transglutaminases.

### 2.3. Effect of pH, Reducing/Oxidising Environment, and Sodium-Dodecyl Sulphate on hTG4 Transamidase Activity

As a next step, we further characterised the transamidase activity of hTG4 under various conditions and searched for regulatory mechanisms that might be responsible for the activation of hTG4, leading to high enzymatic activity. The pH influences the activity of enzymes, either promoting the catalytic mechanism or influencing the solubility of the protein at their isoelectric point. Human TG4 demonstrated slightly higher activities at lower pH values, below pH 7 (Figure 2A). The pH-dependence curve suggests that the isoelectric point of hTG4 is around pH 7.0.

Transglutaminases are also known to be sensitive to the reducing/oxidising environment altering the state of vicinal thiol groups in the protein. Transamidase activity of hTG4 was measured in the presence of various ratios of reduced and oxidised (GSH/GSSG) glutathione (Figure 2B). The results demonstrate that hTG4 prefers the reducing condition to express increased transamidase activity.

As low amounts of SDS potently enhance rat TG4 activity, the effect of increasing SDS concentrations was tested on hTG4 transamidase activity (Figure 2C). Indeed, SDS in the range of 0.1–0.5 mM concentration resulted in an approximately three to four times increase in hTG4 transamidase activity, while higher SDS concentrations were rather inhibitory. This enhancing effect of SDS was not observed in the case of hTG2, where increasing SDS resulted in the inactivation of the enzyme. However, SDS in small concentrations enhanced the activity of hTG4, but still, the activity values were lower than in the hTG2-catalysed reactions. 

### 2.4. Thermal Stability of Recombinant hTG4 Protein

Due to the low activity values, it was suspected that hTG4 could have low stability. To characterise the thermal stability of the protein, nano differential scanning fluorimetry (nanoDSF) was applied. The NanoTemper Prometheus machine determines, without protein labelling, the thermal unfolding by measuring the intrinsic Trp fluorescence while raising temperature under native conditions. Proteins with a higher unfolding transition temperature (Tm, where 50% of the protein is unfolded) are more stable. Applying a serial hTG4 dilution, the average Tm value of hTG4 was 62.6 ± 0.23 °C (Figure 3). The T_onset_, the temperature where the protein starts to unfold, was 56.1 ± 1.34 °C. At the applied protein concentration range, there is no sign of other structural transition, like dissociation. Considering the general human body temperature, hTG4 demonstrates high thermal stability, indicating a stable native structure in physiological conditions. This observation raises the possibility of a still unknown scaffolding function of hTG4.

### 2.5. Human TG4 Does Not Bind GTP

Other transglutaminases are often regulated by guanine nucleotide-binding. The GTP binding property of the enzyme was tested using the fluorescent BODIPY-FL-GTPγS reagent (Figure 4). In the case of a binding event, a protein concentration-dependent fluorescent intensity increase was expected. Due to the lack of changes, recombinant hTG2 was used as a positive control in 1000 nM concentration, which resulted in an approximately four times higher relative fluorescence compared to hTG4, ensuring the correctness of the experimental conditions.

In order to confirm the lack of GTP binding property of hTG4, GTP-agarose and consequent Western blot were used (Figure 4, inlet). The results demonstrate that GTP-agarose resin cannot bind hTG4 protein, while the separation was successful with the positive control hTG2. The two independent results prove that guanine nucleotide does not have a binding site and regulatory effect on hTG4.

### 2.6. Limited Proteolysis of hTG4 by Thrombin or Dispase Is Not a Potential Mechanism for Activation of hTG4

Transglutaminases could also be regulated by limited proteolysis. In order to test the potential activation of hTG4 by thrombin or dispase, first, the protein sequence was checked in silico for cleavage sites, using ExPASy-PeptideCutter online tool (https://web.expasy.org/peptide_cutter/; accessed on 1 July 2021). There is no thrombin recognition site in hTG4 based on the Peptide Cutter, and we confirmed the lack of the effect of thrombin also experimentally (data not shown). Dispase recognition sites are not included in this online tool. So, recombinant His-tagged hTG4 and hTG3 were tested for limited proteolysis experimentally by dispase I and II proteases (Figure 5A). Dispase I shows higher proteolytic activity than dispase II. Western blot analysis using both anti-polyHis/HRP antibody and monoclonal anti-TG4 antibodies demonstrated that dispase I degrades hTG4 wholly and rapidly, while dispase II only decreased the protein level compared to the control suggesting that dispases can cleave hTG4 but do not generate detectable stable fragments. The reason of less observed fragments in the case of anti-polyHis/HRP antibody could be explained by the proteolytic cleavage of *N*-terminal His_6_-tag from the protein fragments. Recombinant hTG3 was tested as a positive control where the anti-TG3 and the anti-polyHis-tag antibodies detected the stable proteolytic *N*-terminal TG3 fragment.

In the AD-293 cell line, a still unknown intracellular protease can activate exogenously expressed human TG5 [29]. We therefore checked whether hTG4 could go through a similar activation mechanism. AD-293 cells were transfected by eukaryotic pTriEx-4 Ek/LIC vector expressing hTG4. Forty-eight hours after transfection, the protein extract contained only the full-size hTG4 indicating no active protease that could modify the protein by limited proteolysis (Figure 5B). While the attempts to find an activator mechanism by limited proteolysis were unsuccessful, in further experiments, we tried to find high-affinity substrates to demonstrate significant catalytic hTG4 activity.

### 2.7. Human TG4 Incorporates Biotin-Pentylamine into Several Proteins in AD-293 Cell Extract

The observed low hTG4 transamidase activity raises the possibility that the application of unpreferred substrates may be responsible for the measured low enzymatic activity. The existence and presence of efficient glutamine-donor protein substrates of hTG4 were tested in AD-293 cell protein extract. AD-293 cells do not contain any transglutaminases and do not possess any transamidase activity (Figure 6A). When the AD-293 cellular extracts were incubated with recombinant hTG4 in the presence of Ca^2+^, several protein bands appeared, having incorporated biotin-pentylamine (BPA). Before the hTG4-dependent BPA labelling, the nuclear (N) and cytoplasm (Cp) fractions of AD-293 cells were separated to increase the efficiency of protein identification. Recombinant hTG4 efficiently incorporated BPA into several proteins in both fractions of the three biological replicates (Figure 6B). To enrich the ratio of BPA labelled proteins, NeutrAvidin-agarose affinity chromatography was applied (Figure 6C), and then the mixtures were sent for mass spectrometry-based (LC-MS/MS) protein identification. NeutrAvidin-agarose bound some cellular proteins in a nonspecific way, therefore we used control samples containing the proteins eluted from the NeutrAvidin-agarose after its incubation with only the untransfected AD-293 cell protein extract.

After subtracting the nonspecifically bound proteins identified in the controls, mass spectrometry analysis of the hTG4-treated samples revealed eight proteins from the cytoplasm and 230 proteins from the nucleus as potential hTG4 substrates or interacting proteins of the substrates. (Appendix A). All analyses were done from three independently prepared cytoplasmic and three nuclear fractions. The detection of a BPA modified peptide fragment of the protein proved the presence of reactive glutamine and the occurrence of the hTG4-mediated BPA-incorporation in the following proteins: in the cytoplasmic fractions, Keratin, type II cytoskeletal 1 and Keratin, type I cytoskeletal 9 got BPA-labelled, but the unmodified forms of these proteins were also identified in the control samples. Their BPA modifications proved that hTG4 can use them as substrates. In the nuclear fraction, only Insulin-like growth factor 2 mRNA-binding protein 1 was BPA-modified. Further study would be needed to confirm and determine the reactive glutamine in the other proteins. Human TG4 was also detected in two out of the three cytoplasmic and in all three nuclear samples. It is conceivable that hTG4 can crosslink itself with an already BPA-labelled substrate using another reactive residue or copurified with a BPA-labelled substrate.

While with the evaluation of the proteomics dataset focused on protein identification only the existence of three substrate proteins was confirmed by BPA modifications, another approach for the data analysis focussing on BPA-modified peptides could also find existing substrate proteins and determine their reactive glutamine residues. Therefore, we filtered the dataset again for the presence of BPA modification in the peptide fragments and removed the previously applied filtering parameters, which are valid for protein determination. We found 105 proteins that contained incorporated BPA. Each peptide’s MS/MS spectrum was verified, and hits, where at least four consequent b or y fragment ions were present, are listed in Table 3 (the 20 peptides are related to 18 proteins because two proteins contain two peptide fragments).

Based on these twenty selected hTG4-modified peptides, we analysed the linear environment of the reactive Gln residues (Figure 7). Unfortunately, it is not possible to deduct an obvious consensus recognition sequence for hTG4. But, hTG4 prefers poly-glutamine tracts and nearby Glu, which could be the result of a transglutaminase-dependent previous deamidation of such poly-glutamine tracts. Leu, Arg, Ser, and Val amino acids also are frequent in the linear surrounding area of the reactive Gln residue.

## 3. Discussion

Human transglutaminases are getting higher attention due to their biological functions and potential medical relevance, but there is a blind spot in our knowledge in regard of human TG4’s biochemical properties and tissue distribution. It has previously been demonstrated that human TGM4 is expressed in several tissues but with a minimum of 200 times lower level than in the prostate [30]. Recently, other studies got a similar conclusion at both RNA and protein levels that human TGM4 expression exists, but it is minimal in nonprostatic tissues [31]. In mice, TG4 is also expressed at the protein level in the vena cava and aortic smooth muscle cells [8]. Focusing on the detection of unique peptides which gives higher reliability, we reanalysed public proteomics data to confirm that TG4 protein is present in various human tissues outside the prostate. Based on our analysis of these real experimental datasets, hTG4 is present in the foetal heart, colon and salivary gland, which could be the source of salivary TG4 [17]. Sandwich ELISA assay demonstrated that TG4 is also present in the blood sera with a still unknown source, similar to the PSA antigen detected at ultralow levels in female sera [32]. Based on the transcription data of TGM4 gene, it is possible that the application of higher sensitivity mass spectrometric devices and technics will lead to the detection of hTG4 protein at a low level in the intestine and its secretory epithelial structures similar to the salivary gland and saliva [17].

The catalytic properties of hTG4 were tested using immobilised *N*,*N*-dimethylated casein and soluble biotin-pentylamine, which are generally accepted substrates for the whole transglutaminase enzyme family. This is one of the most sensitive transglutaminase assays, but the observed hTG4 activity values were low, even in the increased amount of the enzyme. Human TG2 was used as a positive control to prove that the experimental settings were appropriate. We have tested several experimental conditions, but only low SDS concentrations were able to enhance the transamidase activity of hTG4. A similar phenomenon was observed in the case of mouse TG4, which showed increased activity in the presence of approximately 1.5 mM SDS [10], which is not unique, but a rare enzymatic property also known in the case of polyphenol oxidase [33]. Usually, SDS causes denaturation and inactivation of the proteins. However, in the submicellar concentration (<1.5 mM) SDS could promote the formation of an equilibrium intermediate [34], very probably resulting in a loosed, relaxed structure contributing to the active conformation. Further increase of the SDS leads to the complete denaturation of the protein. The observed high thermal stability of hTG4 further supports this theory. SDS may bind to hTG4 by hydrophobic and electrostatic interactions [10], either increasing its solubility or loosening the inactive conformation. Another possibility is that based on the low similarity with other transglutaminases, TG4 could have less Ca^2+^-binding sites. In vitro, isolated rat TG4 is active without exogenously added calcium-ion but demonstrated two high-affinity calcium-binding sites [10]. The difference between calcium-binding properties in the transglutaminase family could be responsible for the generally observed low transamidase activity of hTG4. The SDS activation could mimic the binding of other amphipathic biomolecules regulating hTG4 activity.

Based on our results, hTG4 prefers slightly acidic pH appropriate for the human vaginal conditions, in contrast to mouse TG4, which is active in the alkalic semen, to form a copulatory plug. This hTG4 feature correlates with the change of human sexual strategy leading to the elimination of sperm competition [35]. When hTG4 is present in the intestine, where the pH can drop below 7, this may slightly enhance transglutaminase activity. The absence of alkalic pH preference can also help to silence potential hTG4 activity in blood circulation. Mouse TG4 demonstrated higher activity in oxidising conditions correlating with the lack of reducing agents in the semen. Human TG4 prefers reducing conditions suggesting that its transglutaminase activity could play a role in intracellular processes. We have tested the potential regulatory role of GTP on hTG4, but in contrast to its rat orthologue (sequence similarity is 53.3%), it cannot bind guanine nucleotide [12]. The GTP binding property of hTG2 has been well-described, and the responsible residues have been determined [36]. When the hTG4 sequence was aligned to hTG2, the similarity was low between the two human transglutaminases (42%) [37], and the GTP binding residues were missing in hTG4. We also have observed the lack of fibronectin binding sites contrary to hTG2 (Appendix A). The missing cellular negative GTP regulation of hTG4 can support its effect on tumour invasiveness. In prostate cancer cells, the presence of hTG4 is a negative prognostic marker [31]. Further studies with hTG4 inhibitors would be needed.

The observed low catalytic hTG4 activity could originate from the lack of appropriate enzyme activation. In the transglutaminase family, several members need limited proteolysis for their activation. Physiological thrombin cleavage is essential for FXIII-A in the blood [3]. TG1, TG3, TG5 also need proteolytic cleavages. TG1 could be activated by m-calpain, µ-calpain, dispase and TG3 by cathepsin-l or dispase [38,39,40,41]. In the case of TG5, the processing protease is still not identified, but in AD-293 cells, the exogenously expressed TG5 was present in an inactive full and in an active, proteolytically processed shorter form. Our results suggest that limited proteolysis under similar conditions is probably not involved in the processing and activation of the hTG4 enzyme.

In order to acquire information on the substrate preference of hTG4, we have identified its cellular substrates using LC-MS/MS-based method. Similarly to hTG2 [42], hTG4 can use several proteins in AD-293 cells as glutamine donor substrate. After selecting the most reliable hits from the results, no consensus sequence could be determined. For reference, we checked the TRANSDAB, transglutaminase substrate database (http://genomics.dote.hu/mediawiki/index.php/Main_Page, accessed on 1 July 2021) [43] and found that some of our newly identified hTG4 substrates are substrates for other transglutaminases as well: keratin, type II cytoskeletal 1 for hFXIII-A, filaggrin for hTG1 and hTG3. There are some similarities: NF-kappa-B inhibitor alpha, F-box only protein 2 and Myosin-9 are substrates of hTG2, and Coiled-coil domain-containing protein 126 is a substrate of FXIII-A. Only isoforms of the previous proteins are substrates of hTG4, suggesting that hTG4 has a unique substrate recognition. This is not completely unexpected because mouse TG4 can use F2 (79–259 amino acids) fragment of the seminal vesicle secretion I (SVS I) protein as a glutamine donor while TG2 does not prefer it [44]. Interestingly, the SVS I (Uniprot code: Q6WIZ7_MOUSE) is an amine oxidase with two reactive glutamines (Q232, Q254) and, in the case of its human orthologues (AOC1_HUMAN, C9J2J4_HUMAN), the reactive glutamine residues are changed to Arg or His, respectively.

Then, we have analysed the physiological and pathological roles of the identified substrate proteins. We focused on membrane-localised proteins because hTG4 overexpression contributes to the invasiveness of prostate tumour cells, increasing their adhesion and migration [37]. Based on the Human Protein Atlas, Adhesion G protein-coupled receptor L3 (ADGRL3) is frequently present in breast, prostate and colorectal cancers. Protocadherins, members of the cadherin family, are essential to maintain normal cell–cell interaction and are involved in the epithelial-mesenchymal transition (EMT) [45]. Plexin-A2, as a coreceptor of semaphorins, plays a role in invasive growth and cell migration. Transglutaminase dependent modification of NF-kappa-B inhibitor alpha could promote its degradation and activation of NF-kappa-B, contributing to cell survival similarly to hTG2 [2]. Human TG4 promotes tumour and endothelial cell interaction potentially through influencing the ROCK pathway [19]. Unfortunately, no ROCK1 or ROCK2 interaction partners were found as TG4 substrate. But among the 105 identified TG4 substrates, we can find Rho guanine nucleotide exchange factor 28, which can play a role downstream of the ROCK signalling. The role of these identified substrates in cancer may initiate further studies to reveal their exact role in cancer biology, potentially leading to identifying new cancer therapeutic targets.

While TG4 has the highest polymorphism in the transglutaminase family, it is structurally very stable. We are tempted to speculate that it is on the way to gain a still unknown scaffolding function or it has some substrates contributing to the fine-tuning of biological processes potentially involved in cell proliferation, adhesion, immunological, inflammatory processes on the interfacial surfaces of the intestine, urogenital tract or circulatory vessels. To identify the potential new biological function of hTG4, further functional studies are needed.

## 4. Materials and Methods

All materials were purchased from Sigma-Aldrich (St. Louis, MO, USA) unless otherwise indicated.

### 4.1. Proteomics Database Analysis

“PSM Provenance” table was exported from the MassIVE database (Mass Spectrometry Interactive Virtual Environment; https://massive.ucsd.edu, accessed on 1 July 2021) [26]. The interface of the MassIVE allows to filter the peptides according to their uniqueness by filtering the Matched Genes feature for the peptides. According to that list, the unique and the shared peptides were separated into two tables. The origin of the samples where the peptides were identified was mined out from the names of the files or based on the dataset identifier. In the case of 76 out of 3499 peptides, their origins were not available. After clicking on the observed TG4 peptides in the PeptideAtlas (http://www.peptideatlas.org, accessed on 1 July 2021) [27] under the “Sequence Motifs” feature, the data from the “Observed in Experiments” part was collected. Whether a given peptide was unique for hTG4 was checked in each case using NCBI Standard Protein BLAST online tool. The tissues were collected from the Experiment Names, which were always referring to the sample origin. In ProteomicsDB (https://www.proteomicsdb.org, accessed on 1 July 2021) [28], under the “Peptides/MSMS” tab, 49 unique and one shared hTG4 peptide can be found, which were identified in the datasets of the ProteomicsDB. Clicking on the peptide sequences, the “Peptide Details” appear, which were exported to Excel (Microsoft, Redmond, WA, USA) in the case of all 50 peptides. Based on the project, experiment names, or file names the origin of the sample can be identified. In the case of publicly available datasets, by clicking on the experiment, the sample mapping can be reached.

### 4.2. Production of Recombinant Human TG4

The coding part of TG4 cDNA (Uniprot code: P49221) was subcloned into pBH4 [46] bacterial expression vector using NdeI and XhoI restriction sites in the frame after an *N*-terminal His_6_ tag from pDNR-LIB vector (IMAGE clone: 3950865). The appropriate insertion was checked by restriction analysis and sequencing (Eurofins Genomics Germany GmbH, Ebersberg, Germany). The human recombinant TG4 protein was expressed in *E. coli* BL21 (DE3)pLysS bacterial cells using Terrific Broth (TB). An overnight culture was inoculated in 1:20 ratio for large scale expression and the cells were grown at 30 °C until reaching OD600 = 0.6. Protein expression was induced at 18 °C, in the presence of 0.2 mM isopropyl-β-d-thiogalactoside for 16–20 h. Cells were harvested by centrifugation (6000× *g*, 10 min, 4 °C) and resuspended in lysis buffer (Talon binding buffer: 50 mM phosphate buffer, 300 mM NaCl, 20 mM Imidazole, pH 7.6, containing 1 mM PMSF and 10% glycerol). After sonication and centrifugation (20,000× *g*, 1 h, 4 °C) the supernatant was mixed with Talon Metal Affinity Resin (Takara Bio USA, Inc., San Jose, CA, USA). After 1 h incubation at 4 °C, the beads were washed using Talon buffer containing 20 mM imidazole. Proteins were eluted using 150 mM Imidazole containing Talon buffer, and the buffer was exchanged to Buffer A (50 mM Tris-HCl, pH 7.6, 0.5 mM EDTA, 15% glycerol) using Amicon Ultra 50 K centrifugal concentrator unit (Merck, Darmstadt, Germany). For ion exchange chromatography HiTrapQ HP Column (GE Amersham, Buckinghamshire, UK), to create gradient Buffer B (Buffer A containing 1 M NaCl) were used. The protein concentrations were always determined using Bio-Rad Protein Assay Dye Reagent Concentrate (Bio-Rad, Hercules, CA, USA). The purity of rhTG4 (better than 90%) was checked by Coomassie staining and Western blot using monoclonal anti-TG4 antibody (mab0113-P, Covalab S.A.S., Bron, France).

### 4.3. Transamidase Activity Assay

The classical microtitre plate assay [47] was used to measure the hTG4 transamidase activity with small modifications. Based on the enzyme concentration dependence, 2.5 µg hTG4 was chosen for further activity measurements. The transglutaminase reaction was usually performed in the presence of 0.5 mM biotin-pentylamine substrate (B002, Zedira; Darmstadt, Germany,) and 10 mM DTT and 5 mM Ca^2+^ at 37 °C for 45 min. Extravidin alkaline phosphatase was used in 1/2500 dilution. The colour change during phosphatase reaction was followed at 405 nm using Synergy H1 device (BioTek Instruments, Inc, Winooski, VT, USA). The transglutaminase reaction rate was calculated from the initial slopes of the kinetic curves gained from the phosphatase reaction in mAbs change per minute. To determine the pH-dependence of hTG4, the assay buffer was 0.1 M Tris-HCl buffer in the case of pH 8.5 and 7.5, but in the case of pH 7, 6.5, 6 and 5.5 50 mM MOPS buffer was used. To set up various reducing or oxidising conditions the reaction mixture was supplemented with 2.4 mM GSSG and 0, 0.24, 0.45, 1.25, or 2.4 mM GSH [28]. The ratio of [GSH]^2^ to [GSSG] was calculated from their final molar concentrations. The graphs were prepared and edited using Graph Pad Prism 8 (GraphPad Software, San Diego, CA, USA).

### 4.4. Nucleotide-Binding Assays

The nucleotide-binding of hTG4 was tested using BODIPY-FL-GTPγS (Invitrogen, Carlsbad, CA, USA) [46] and GTP-agarose [48] based on previously described methods with some modifications. The protein-bound BODIPY-FL-GTPγS possesses higher fluorescence measured by Synergy H1 plate reader (Ex/Em:485/520;). The experiment was performed twice using three parallels in Nunc black microplates. As a positive control, 1000 nM recombinant hTG2 was used. For the GTP agarose pull-down assay, purified 5 µg recombinant hTG4 or hTG2 were used. Bound transglutaminases were eluted by adding 2× denaturation buffer and 10 min of boiling. Western blot was developed using anti-polyHis/HRP antibody (A7058) in 1/15,000 dilution.

### 4.5. Differential Scanning Fluorimetry

The thermal stability of human TG4 was tested using differential scanning fluorimetry. The Prometheus NT.48 device makes possible the label-free detection of thermal unfolding measuring changes of intrinsic fluorescence of the protein (tyrosine and tryptophan) during increasing the temperature from 20 to 95 °C. The hTG4 protein was diluted in 20 mM Tris-HCl buffer, pH 7.2 containing 150 mM NaCl, 1 mM EDTA, 1 mM DTT.

### 4.6. Dispase Digestion

To test the proteolytic stability, the reaction mixture contained 1 µg TG4 orTG3 and 1 µg dispase I or II and 5 mM Ca^2+^ in 0.1 M Tris-HCl buffer, pH 8.5. After 1 h incubation at 37 °C, the reaction was stopped by the addition of 6× denaturation buffer (300 mM Tris-HCl pH 6.8, 12 (*m*/*v*)% SDS, 20 (*v*/*v*)% glycerol and 0.43 M β-mercaptoethanol). The reaction mixture was analysed by Western blot, applying polyclonal antihuman epidermal transglutaminase (TG3) antibody (A015, Zedira, 1/2000), monoclonal anti-TG4 antibody (Covalab; 1/2500) and monoclonal anti-polyHis/HRP antibody (1/20,000).

### 4.7. Exogenous Expression of hTG4 in AD-293 Cells

AD-293 cells were maintained in Dulbecco’s modified Eagle’s medium completed by 1% PS, 1% pyruvate, 2% glutamine and 10% FBS (Thermo Fisher, Waltham, MA, USA). Cells were plated on a 6 well plate. After reaching the 70% confluence, the FBS was lowered to 2%, and antibiotics were omitted. The pTriEx-4 Ek/LIC hTG4 vector was constructed using pTriEx™-4 Ek/LIC Vector Kit-Novagen (Merck) based on the manufacturer instruction. The hTG4 cDNA containing pBH4 vector was used as a template with the gacgacgacaagatgagaattcagaccgaaaacctgtacttcc and gaggagaagcccggtccaactcagcttcctttc oligonucleotides. The transfection was performed based on the manufacturer’s instructions using Lipofectamine 2000 (Thermo). After 3 h, the transfection medium was replaced with a culture medium. The cells were collected 48 h after transfection using RIPA buffer (50 mM Tris-HCl pH 7.5, 150 mM NaCl, 1 mM EDTA, 1 mM DTT, 0.5 (*v*/*v*)% NP-40, 10 (*v*/*v*)% glycerol and 1 mM PMSF and 1% protease inhibitor cocktail).

### 4.8. Human TG4 Substrate Search

AD-293 cells were cultured until 90% confluency and collected by trypsinisation. After centrifugation at 1000× *g* rpm for 10 min, the pellet was washed twice with ice-cold PBS. First, to prove that TG4 can use glutamine donor proteins from AD-293 cells, the cell pellet was resuspended in RIPA buffer and sonicated. The cell debris was removed by short centrifugation (5 min, 18,000× *g*, 4 °C). 10 µg AD-293 cell extracts were incubated for 1 h, at 37 °C together with 0.2 mM BPA, with or without 4.2 µg rhTG4 in 100 mM Tris-HCl buffer, pH 8.5 containing 10 mM DTT, 1 mM PMSF, 1% protease inhibitor cocktail, and 10 mM EDTA or 5 mM CaCl_2_,. The reaction was stopped by adding 6× denaturation buffer (300 mM Tris-HCl pH 6.8, 12 (*m*/*v*)% SDS, 20 (*v*/*v*)% glycerol and 0.43 M β-mercaptoethanol). Incorporated BPA was analysed by Western blot using Streptavidin/HRP (405210, BioLegend San Diego, CA, USA, 1/2500).

The incorporation reaction was repeated three times to identify the BPA-labelled hTG4 substrate proteins on separated cytoplasmic (Cp) and nuclear (N) protein fractions of AD-293 cells. For the separation, Nuclei Isolation Kit: Nuclei EZ Prep (26149, Sigma) was used according to the manufacturer protocol. Then, 100 µg cytoplasmic or nuclear fraction of AD-293 cell extract were incubated in 0.1 M Tris-HCl buffer, pH 8.5 containing 5 mM CaCl_2_, 10 mM DTT, 200 µM BPA and 3 µg recombinant hTG4 for 2 h, at 37 °C. The excess BPA was removed by dialysis against 0.1 M Tris HCl buffer, pH 8.5 using Spectra/Por 1 Dialysis Membrane Standard RC Tubing (MWCO 6–8 kD). BPA-modified proteins were isolated and enriched using High Capacity Neutravidin Agarose Resin (Thermo). After equilibration (0.1 M Tris-HCl buffer, pH 8.5), the resin was incubated with the dialysed reaction mixture for 1 h, at 4 °C under continuous rotation. The beads were washed three times, and the labelled proteins were eluted by the addition of 6× denaturation buffer, and subsequent boiling for 15 min. The supernatant was used for further analysis. In the case of control samples, 100 µg of AD-293 protein extract was applied for incorporation, treated similarly to the other samples.

The eluted proteins were loaded on 10% polyacrylamide gel, but they were only run into the separating gel. After Coomassie staining, the protein bands were excised from the gel, and the dye was removed by washing with nanopure water. After reduction with DTT, and alkylation using iodoacetamide, the proteins were digested with trypsin overnight, at 37 °C. The extracted peptides were dried in Speed-Vac then reconstituted in 10 µL 1% formic acid. Half of the sample was used for the analysis. First, the salt and accompanying buffer components were removed and the sample was enriched with ACQUITY UPLC Symmetry C18 (Waters Milford, MA, USA) desalting column. Then the peptides were separated using Acclaim PepMap RSLC (Thermo) column (300 nL/min flow rate, 180 min water-acetonitrile gradient). Next, the samples were analysed with Easy nLC1200 (Thermo) nanoUPCL-Orbitrap Fusion (Thermo) MS/MS system. Data collection was performed in data-dependent acquisition mode. The Orbitrap analyser scanned the samples with 6000 mass resolution between 350–1600 *m*/*z* range and the 14 most intensive peaks were used for the further fragmentation, when CID (35%) was applied and the product ions were scanned in the ion trap (MS/MS). The mass spectrometry analyses were carried out at the Proteomics Core Facility, Department of Biochemistry and Molecular Biology, University of Debrecen.

Protein identification based on the peptide sequences (MS/MS spectra) was done by MaxQuant 1.6.2.10. search engine [49] using SwissProt/Uniprot database. The results were imported into the Scaffold 4.8.9 program (Proteome Software, Inc., Portland, OR, USA), and for the protein identification, the following settings were applied: protein threshold 1.0% False Discovery Rate (FDR), the minimum necessary identified peptide number 3 for each protein, and peptide threshold 0.1% FDR. In the case of determination of BPA labelled proteins, non-FDR filtering mode and one peptide per protein criteria were used. In addition, the MS/MS spectrum of each peptide was verified by the presence of at least four consequent b or y fragment ions. The data were reposited to ProteomeXchange Consortium [26] via MassIVE with the PXD028174 data identifier (link: http://proteomecentral.proteomexchange.org/cgi/GetDataset?ID=PXD028174, accessed on 28 August 2021). 

### 4.9. Western Blot

The Western blot was performed based on the general protocol. SDS-PAGE with 10% gel was performed and followed by a semidry blotting method using PVDF membrane (Immobilon-P membrane, 0.45 µm) using Bio-Rad Trans-Blot SD Semi-Dry transfer cell device. 5 (*m*/*v*)% low-fat milk powder in TTBS was used for blocking, for 1 h at room temperature or overnight at 4 °C. Primary antibodies were applied in 0.5 (*m*/*v*)% low-fat milk powder containing TTBS for 1 h at room temperature or overnight at 4 °C. After washing four times for 8 min with TTBS, the secondary antibody was added in TTBS containing 0.5 (*m*/*v*)% low-fat milk powder. The following antibodies were used in the study: Monoclonal Transglutaminase-4 antibody (1C6) (Covalab) (1/2500), Polyclonal antibody to human epidermal transglutaminase (TG3) (Zedira) (1/2000), Monoclonal anti-polyHis/HRP antibody (Sigma) (1/20,000), Streptavidin/HRP (1/2500) (BioLegend), Goat-anti-mouse IgG (R-05071-500, Advansta, San Jose, CA, USA) (1/10,000) and Goat-anti-rabbit IgG (H + L), HRP conjugate (R-05072-500, Advansta) (1/10,000).

## Figures and Tables

**Figure 1 ijms-22-12448-f001:**
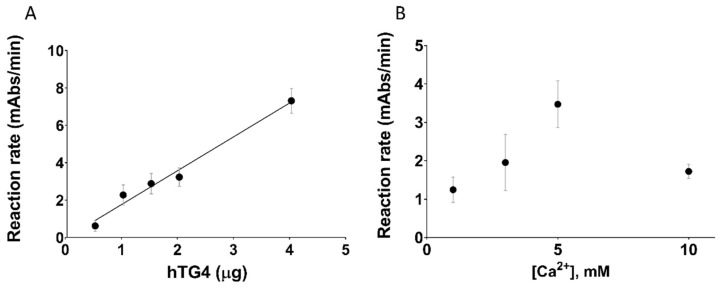
**Transamidase activity of human TG4.** (**A**) The effect of increasing hTG4 enzyme amount on the transamidase activity was measured in the presence of 0.5, 1, 1.5, 2 and 4 µg hTG4 protein and 5 mM Ca^2+^, at pH 8.5. (**B**) The calcium-dependence of hTG4 transamidase activity was tested between 1–10 mM calcium concentration range in the presence of 2.5 µg enzyme at pH 8.5. All data are represented as means with ± SD using Graph Pad Prism 8 from three independent measurements done in duplicates.

**Figure 2 ijms-22-12448-f002:**
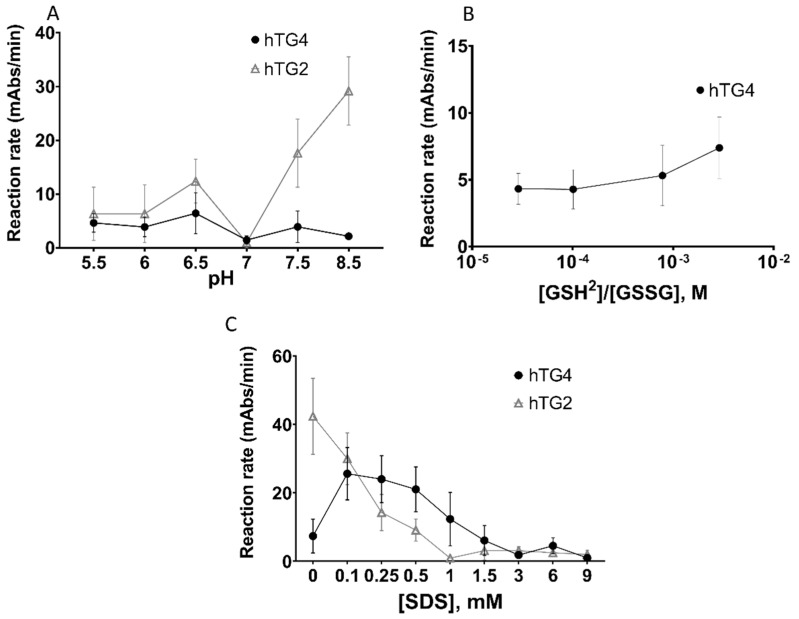
**Effect of pH, reducing or oxidising environment, and sodium-dodecyl sulphate on hTG4 transamidase activity.** The pH dependence of hTG4 and hTG2 transamidase activities between 5.5 and 8.5 pH (**A**). Effect of redox conditions on hTG4 transamidase activity (**B**). To observe the effect of the reducing conditions, 2.4 mM GSSG was combined with 0, 0.24, 0.45, 1.25, or 2.4 mM GSH in the microtitre plate reaction mixture [28]. The ratio of [GSH]^2^ to [GSSG] was calculated from their molar concentrations. Human TG4 represented 8.13 ± 1.65 mAbs/min activity in the presence of 2.4 mM GSH and 0 mM of GSSG, which could not be shown on the chart. Transamidase activities of hTG4 and hTG2 in the presence of various SDS concentrations were tested (**C**). In the experiments, hTG4 amount was 2.5 µg, and hTG2 amount was 0.5 µg. All data are represented as means with ± SD using Graph Pad Prism 8 from two or three independent measurements done in triplicates.

**Figure 3 ijms-22-12448-f003:**
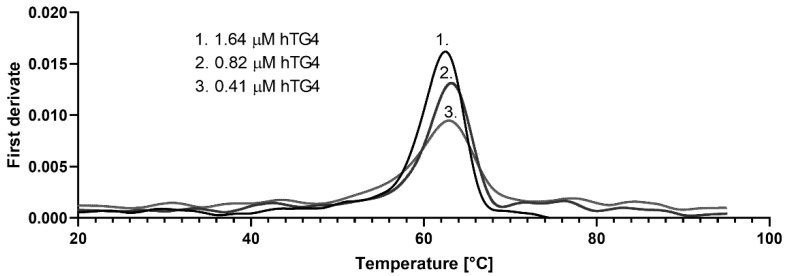
**Thermal stability of recombinant hTG4 protein.** Nano differential scanning fluorimetry was used to determine Tm values, demonstrating the melting scan by the first derivate versus temperature applying 1.64, 0.82 and 0.41 µM hTG4 protein.

**Figure 4 ijms-22-12448-f004:**
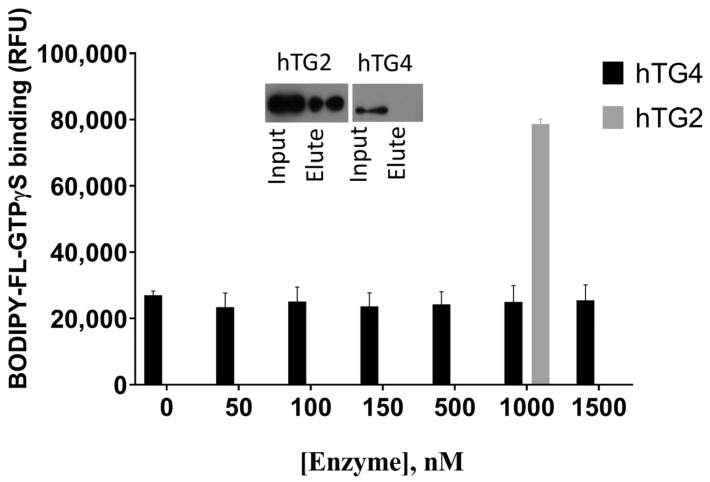
**Human TG4 does not bind BODIPY-GTPγS or GTP.** Increasing concentrations of hTG4 (50, 100, 150, 500, 1000 or 1500 nM) were incubated with 500 nM BODIPY-GTPγS for 5 min, and then the fluorescence intensity was measured (Ex/Em: 485/520 nm). Data are presented as means with ± SD from two separate experiments done in triplicates. As a positive control, 1000 nM hTG2 was applied, resulting in 78,629 ± 153,160 RFU. The inlet shows the binding property of hTG2 and hTG4 to GTP-agarose resin. Representative picture of Western blots after SDS-PAGE from two independent experiments using monoclonal anti-polyHis/HRP antibody.

**Figure 5 ijms-22-12448-f005:**
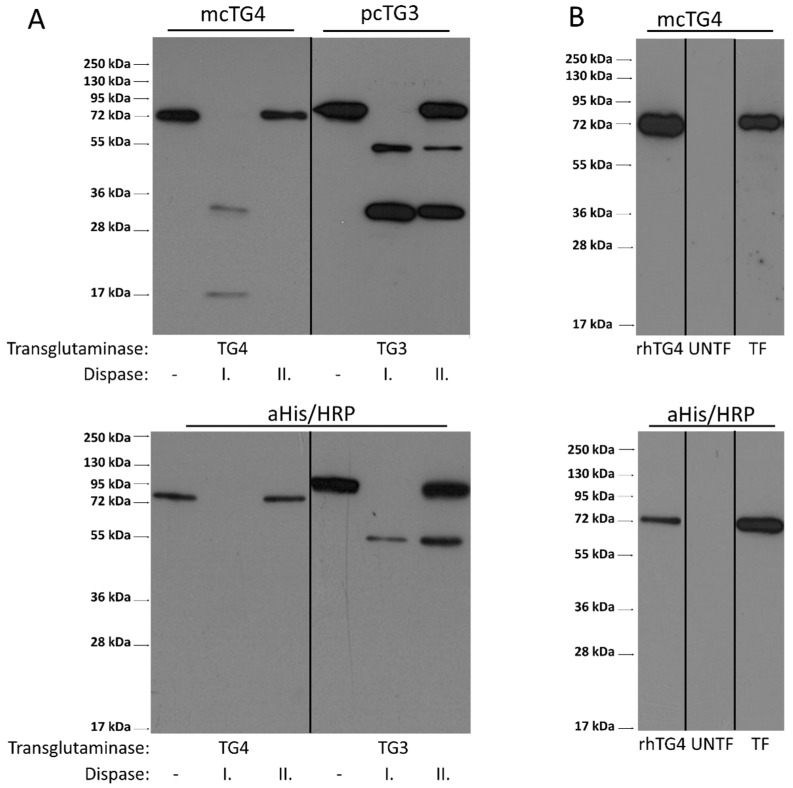
**Limited proteolysis of hTG4 protein does not occur upon dispase digestion or its expression in AD-293 cells.** (**A**) The effects of dispase I and II on hTG4 protein were examined by Western blot analysis. Recombinant hTG3 or hTG4 were incubated for 60 min at 37 °C, in the presence of 5 mM Ca^2+^ alone or with the same amount of either dispase I or dispase II. From each reaction, 0.3 µg transglutaminase protein was tested by Western blot using monoclonal anti-TG4 (mcTG4) or polyclonal anti-TG3 (pcTG3) antibodies (upper part) and anti-polyHis/HRP antibody (aHis/HRP). Representative gel pictures of two independent experiments are presented. (**B**) AD-293 cells express exogenous hTG4 in full-length form without significant limited proteolysis revealed by Western blot. 0.3 µg recombinant hTG4 (rhTG4) was used as positive control while 10 µg protein extract of untransfected (UNTF) and transfected AD-293 cells (TF) were examined. Representative picture from three independent experiments. The vertical lines on the blots are labelling the place of the membrane cuttings.

**Figure 6 ijms-22-12448-f006:**
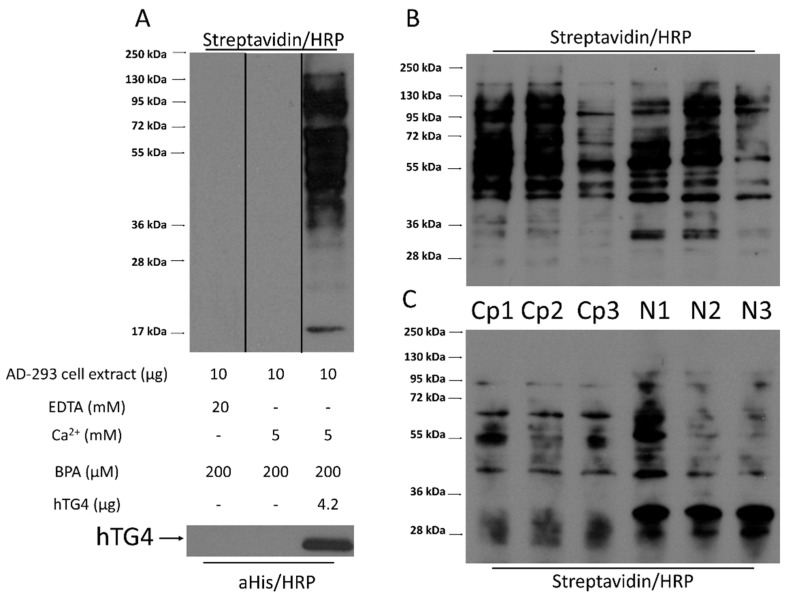
**Verification of biotin-pentylamine (BPA) modified proteins in AD-293 cell extract upon incubation with hTG4.** (**A**) BPA incorporation via hTG4 transamidase activity into AD-293 cell extract examined by Western blot, using Streptavidin/HRP for detection. The bottom part of the Western blot picture shows the presence of hTG4 in the sample, detected using anti-polyHis/HRP. The vertical lines on the blot indicate the place of the membrane cuttings. Representative picture from two independent experiments. The reaction conditions are summarised in the picture. BPA incorporation into AD-293 cytoplasmic (Cp) and nuclear (N) fractions directly after the enzyme reaction (**B**) and after the separation of the BPA containing substrates by NeutrAvidine Agarose (**C**). Western blots using Streptavidin/HRP for detection.

**Figure 7 ijms-22-12448-f007:**
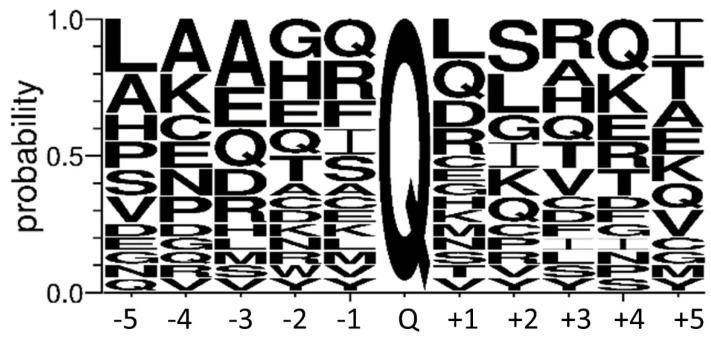
**Probability of the amino acid residues between the −5 and +5 position to the reactive glutamine.** Sequence logo created using the 20 most reliable peptide sequences containing the detected BPA-modified Gln (Q) with its surrounding amino acid residues in both directions. The size of the amino acid letter represents the probability of the given amino acids in the substrate recognition region (created by WebLogo 3.8.4. online tool; http://weblogo.threeplusone.com/, accessed on 1 July 2021).

**Table 1 ijms-22-12448-t001:** Human TG4 expressing tissues based on detected unique peptides in MassIVE, PeptideAtlas or ProteomicsDB databases.

Tissue	MassIVE	PeptideAtlas	ProteomicsDB
Colon	++	++	+
Heart	+	++	++
Prostate	++	++	++
Salivary gland	−	−	++
Seminal plasma	++	++	++
Seminal vesicle	++	++	++
Spermatozoon	++	++	−
Spleen	−	+	++
Urinary bladder	++	++	++

−: not detected +: detected only with non-unique peptides ++: detected with unique peptides.

**Table 2 ijms-22-12448-t002:** **Comparison of transamidase activity of human recombinant transglutaminases.** Transamidase activities of TG1, TG2, TG3, TG4 and TG7 were collected from the Zedira GmBH (Darmstadt, Germany) website (www.zedira.com, accessed on 1 July 2021), from the products datasheets. The activity values were uniformly determined by measuring the incorporation of the dansyl-cadaverine into *N*,*N*-dimethylated casein.

Human Transglutaminase	Article-No.	Transamidase Activity (U/mg)	Expression Host
TG1	T009	~2500	*E. coli*
TG2	T001	~750	*E. coli*
TG3	T024	~1000	Insect cells
TG4	T042	~30	*E. coli*
TG7	T011	~1000	*E. coli*

**Table 3 ijms-22-12448-t003:** Potential hTG4 substrate proteins containing BPA modified peptides detected with at least four consequent fragment ion series. Reactive glutamines are labelled by underlined and bold letters.

	Accession	Identified Protein	Cellular Localisation	Total Spectrum Count	Reactive Q ± 5 AA
1.	B4GT5	Beta-1,4-galactosyltranseferase 5	Golgi apparatus	13	AQVYE**Q**VLRSA
2.	AGRL3	Adhesion G protein-coupled receptor L3	Cell membrane	2	HGSTI**Q**LSANT
3.	ENO4	Enolase 4	Cytosol	4	SKRGQ**Q**QITGK
4.	CMYA5	Cardiomyopathy-associated protein 5	Cytoplasm, Nucleus	4	SDLGR**Q**SGSIG
5.	CHD1	Chromodomain-helicase-DNA-binding protein 1	Cytoplasm, Nucleus	1	LEHTR**Q**CLIKI
6.	SAM15	Sterile alpha motif domain-containing protein 15	n.a.	1	VPEEM**Q**RKATE
7.	GRIN3	G protein-regulated inducer of neurite outgrowth 3	Plasma membrane	2	LPAQR**Q**MSRFK
8.	K2C79	Keratin, type II cytoskeletal 79	Cytoskeleton, Cytosole, Extracellular exosome	1	AEAWY**Q**TKYEE
9.	PCDH17	Protocadherin-17	Cell membrane	1	HNAKC**Q**LSLEV
10.	F10A5	Putative protein FAM 10A5	Cytoplasm	1	PNAAI**Q**DCDRA
11.	PCDHB3	Protocadherin beta-3	Cell membrane	1	NKQHF**Q**LSHQT
12.	STAB2	Stabilin-2	Cytoplasm, Cell membrane	1	VARCS**Q**KGTKV
13.	PLXA2	Plexin-A2	Cell membrane	1	AVDGK**Q**DYFPT
14.	F155A	Transmembrane protein FAM155A	Membrane	1	DKEHQ**Q**QQRQQ
					QRQQQ**Q**QQQQQ
15.	ALKB8	Alkylated DNA repair protein alkB homolog 8	Nucleus, Cytoplasm	1	GCDRS**Q**NLVDI
16.	IL6RA	Interleukin-6 receptor subunit alpha	Cell membrane, Secreted	1	PAEDF**Q**EPCQY
17.	ZN363	RING finger and CHY zinc finger domain-containing protein 1	Cytoplasm, Nucleus	1	LAMNL**Q**GRHKC
18.	FAM184A	Protein FAM184A	ECM	1	LCAEA**Q**HVQRI
					EAQHV**Q**RIVTM

## Data Availability

MS data was deposited to ProteomeXchange Consortium [26] via MassIVE with the dataset identifier PXD028174 (MSV000088030). Further data will be available upon request from the corresponding author.

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
