# Peer review of "Biochemical Characterisation of Human Transglutaminase 4"

_ijms, 2021, doi:10.3390/ijms222212448_

Round 1

Reviewer 1 Report

The article is suitable for publication.

Author Response

Comments and Suggestions for Authors:

“The article is suitable for publication.”

Answer: We thank the reviewer’s time to check our manuscript.

Reviewer 2 Report

The current manuscript presents the detailed biochemical characterization of human transglutaminase 4 (hTG4) enzyme. Considering the association of hTG4 with development of autoimmune and tumor disease, it is of crucial importance to understand the biochemical properties of this enzyme. The authors have designed a systematic outline for the detailed characterization, where they have studied the amidase activity of TG4 under various conditions and have also tested its binding property towards GTP. In addition to this, authors have also identified the cellular substrate for hTG4. Although, the data looks convincing, and the conclusions drawn are reasonable , the following concerns should be addressed to improve the quality of the manuscript:

  1. The authors have reported that the transamidase activity of hTG4 is significantly lower than thehTG2 and follows a liner dose dependent response. Although the tested concentration range 0.5ug-4ug appears to be broad but authors have only shown the effect of three concentrations (fig.1 A). It is advised that authors should incorporate more dose dependent data points in Figure 1A to postulate that it indeed follows a linear correlation. Authors should also indicate the pH at which experiment was performed in the figure legend (fig.1), as they later studied the pH dependence on hTG4 activity (fig.2A).

  1. In fig. 1B authors have demonstrated the effect of Ca2+ ion on the transamidase activity if hTG4. With increasing concentration of Ca2+ the activity of hTG4 shows an increase in activity upto 5mM concentration. This is little confusing as at ~1mM Ca2+ concentration the activity of hTG4 seems to be reduced by more than 50 % (~1.2 mAbs/min) compared to the value in absence of Ca2+ (~3mAbs/min) reported in fig. 1A for the similar concentration of hTG4(~2.4ug). Authors should clarify this anomaly as in the present form it is confusing. Also, it has been stated that at 10mM Ca2+ concentration reduced activity of hTG4 was observed and authors have attributed this to plausible precipitation of protein. Did authors observe the visible precipitation of protein? If yes, then it should not be called as inhibition of enzymatic activity as there will be less protein available in solution to begin with which will result in the lesser activity.

  1. In fig. 2B author have demonstrated the effect of redox conditions on hTG4 transamidase activity and concluded that the reducing conditions favor the transamidase activity of hTG4. However, it is unclear why author chose to present the reaction rate in % unit here, whereas for all the other figures they have used the mAbs/min unit. Authors are advised to clearly state how this % reaction rate was calculated and for fair comparison with respect to effect of pH and SDS on hTg4 activity, effect of redox conditions should also be shown in the same unit which mAbs/min.

Authors are also advised to rationalize the increased activity of hTG4 observed at 0.5mM SDS concentration. Although, authors have indicated in the discussion section that at sub micellar concentration SDS could introduce an intermediate structure which could be a more active conformation. It is strongly recommended to perform a dose dependence experiment for the effect of SDS on transamidase activity of hTG4 in the 0-0.5 mM concentration range. At present it is not clear how SDS is facilitating transamidase activity at lower concentration (as there is only one datapoint for lower concentration of SDS in fig. 2B) whereas inhibiting the same at higher concentration.

  1. It is recommended to compare the Tm observed for hTG4 with its rat analog and with TG2. Considering its high stability (Tm=~62.6) and low activity it would be interesting to see the same correlation for the other two protein.
  2. Figure 5A. upper panel looks convincing as fragments of hTg4 can be seen in presence of dispase 1 through mcTG4 staining but it is unclear why those bands are not visible in lower panel for aHis/HRP staining?

Author Response

We thank the Reviewer for the constructive comments, which helped us to improve our manuscript.

Comments and Answers:

Comment 1: “The authors have reported that the transamidase activity of hTG4 is significantly lower than thehTG2 and follows a liner dose dependent response. Although the tested concentration range 0.5ug-4ug appears to be broad but authors have only shown the effect of three concentrations (fig.1 A). It is advised that authors should incorporate more dose dependent data points in Figure 1A to postulate that it indeed follows a linear correlation. Authors should also indicate the pH at which experiment was performed in the figure legend (fig.1), as they later studied the pH dependence on hTG4 activity (fig.2A).”

Answer: Based on the reviewer’s advice, we added two new data points to Fig. 1A, and we also indicated the pH value in the figure legend as it was requested. (see page 4 line 125-129)

Comment 2:” In fig. 1B authors have demonstrated the effect of Ca2+ ion on the transamidase activity if hTG4. With increasing concentration of Ca2+ the activity of hTG4 shows an increase in activity upto 5mM concentration. This is little confusing as at ~1mM Ca2+ concentration the activity of hTG4 seems to be reduced by more than 50 % (~1.2 mAbs/min) compared to the value in absence of Ca2+ (~3mAbs/min) reported in fig. 1A for the similar concentration of hTG4(~2.4ug). Authors should clarify this anomaly as in the present form it is confusing. Also, it has been stated that at 10mM Ca2+ concentration reduced activity of hTG4 was observed and authors have attributed this to plausible precipitation of protein. Did authors observe the visible precipitation of protein? If yes, then it should not be called as inhibition of enzymatic activity as there will be less protein available in solution to begin with which will result in the lesser activity.”

Answer: During the enzyme-dependence measurement of TG4 transamidase activity (Fig. 1A), 5 mM calcium concentration was applied. We added this information to the figure legend (fig.1A.; page 4 lane 127). Then, we have determined the optimal Ca-ion concentration in Fig 1B (5 mM), and it was applied for further experiments. We have not observed visible protein precipitation, so we corrected the text and replaced the word “precipitation” with the word “aggregation” in the manuscript (page 3 line 122).

Comment 3: “In fig. 2B author have demonstrated the effect of redox conditions on hTG4 transamidase activity and concluded that the reducing conditions favor the transamidase activity of hTG4. However, it is unclear why author chose to present the reaction rate in % unit here, whereas for all the other figures they have used the mAbs/min unit. Authors are advised to clearly state how this % reaction rate was calculated and for fair comparison with respect to effect of pH and SDS on hTg4 activity, effect of redox conditions should also be shown in the same unit which mAbs/min.”

Answer: Thank you for raising our attention to this defect. Originally, in Fig 2B. the data was presented in % of enzyme activity in the presence of only 2.4 mM GSH and absence of GSSG (0 mM, which could not be used in the denominator). Based on your request, we converted back the enzyme activity on the figure into mAbs/min units. The activity of hTG4 in the presence of only 2.4 mM GSH was inserted in the figure legend (page 6, line 170-171).

Comment 4: “Authors are also advised to rationalize the increased activity of hTG4 observed at 0.5mM SDS concentration. Although, authors have indicated in the discussion section that at sub micellar concentration SDS could introduce an intermediate structure which could be a more active conformation. It is strongly recommended to perform a dose dependence experiment for the effect of SDS on transamidase activity of hTG4 in the 0-0.5 mM concentration range. At present it is not clear how SDS is facilitating transamidase activity at lower concentration (as there is only one datapoint for lower concentration of SDS in fig. 2B) whereas inhibiting the same at higher concentration.”

Answer: Thank you for this suggestion. We repeated the experiment with lower SDS concentrations and added these data points to the figure, which made it more clear, that SDS concentrations between 0.1-0.5 mM range can increase hTG4 transamidase activity (page 5, line 157). This range is still in the submicellar concentration, as it was already stated and discussed in the manuscript.

Comment 5: “It is recommended to compare the Tm observed for hTG4 with its rat analog and with TG2. Considering its high stability (Tm=~62.6) and low activity it would be interesting to see the same correlation for the other two protein.”

Answer: We focused on human TG4, and mouse or rat TG4 proteins are not available for us and there is no similar data in the literature for rodents TG4. The characterization of human TG2 thermostability and its modification by Ca and GTP is the subject of our other actual study. Tm of hTG2 is around 50oC (unpublished result).

Comment 6: “Figure 5A. upper panel looks convincing as fragments of hTg4 can be seen in presence of dispase 1 through mcTG4 staining but it is unclear why those bands are not visible in lower panel for aHis/HRP staining?”

Answer: The recombinant enzymes have a His6 -tag on the N-terminus, which is most probably removed by the cleavage from the protein by the dispase digestion. This could be the explanation for the disappearance of hTG4 fragments using aHis/HRP, while the monoclonal TG4 antibody can detect the fragments. We made it more clear in the manuscript (page 8. line 227-229).

Reviewer 3 Report

I read “Biochemical characterisation of human transglutaminase 4” The authors analysed proteomics databases and found that TG4 protein is present in human tissues beyond the prostate. 

First it has been demonstrated that human TG4 has low transamidase activity which prefers slightly acidic pH and a reducing environment. It is enhanced by submicellar concentrations of SDS suggesting that membrane proximity is an important regulatory event. From results of interest human TG4 does not bind GTP.

In order to characterize the involvement of TGase activity, several potential human TG4 glutamine donor substrates were analyzed with mass spectrometry. These results suggest that human TG4 could become an anticancer therapeutic target.

The paper is well written and can be of interest in TG issue.

Considering the sentence on TG4 localization, the authors should also explain the manner by which an intracellular enzyme involved in peptide metabolism inside the cells, may be active affecting antigen presentation.

I’m curious about the possibility to enlarge these observations. The statement on the lack of interaction between GTP and TG4, could be characterized by evaluating the action of possible inhibitors of the reaction. A sentence on further study should be added.

Minor comment:

In Figures 2 the reaction rate in y-axis, is reported as mAbs/min. The evaluation of enzyme activity should also performed by comparison with a standard curve using hTG4.

Author Response

We thank the Reviewer for the comments, which helped us to improve our manuscript.

Comments and Suggestions for Authors

Comment 1: “Considering the sentence on TG4 localization, the authors should also explain the manner by which an intracellular enzyme involved in peptide metabolism inside the cells, may be active affecting antigen presentation.”

Answer: We have modified the introduction based on the reviewer comment to make it more clear (page 2 line 56-57). TG4 is secreted into the semen and may act on antigenic surface proteins by its transglutaminase activity. There is no published data about the involvement of hTG4 in intracellular antigen processing.

Comment 2: I’m curious about the possibility to enlarge these observations. The statement on the lack of interaction between GTP and TG4, could be characterized by evaluating the action of possible inhibitors of the reaction. A sentence on further study should be added.

Answer: We have added to the discussion the requested sentence (page 13, line 381-382).

Minor Comment:  “In Figures 2 the reaction rate in y-axis, is reported as mAbs/min. The evaluation of enzyme activity should also performed by comparison with a standard curve using hTG4.”

Answer: Based on the enzyme concentration dependence (Fig. 1A.) 2.5 µg hTG4 was used in all further transglutaminase activity assays (page 15 line 482). We did not use the enzyme concentration when the unit of the activity was calculated because it was constant. We made it more clear in the figure legend (page 6, line 172-174). Using microtiter plate activity assay, we can not demonstrate the exact number of modified substrates. The activity value correlates with the enzyme activity, and the application of concentration in the activity unit would not add more meaning.